# Species and Fatty Acid Diversity of *Desmodesmus* (Chlorophyta) in a Local Japanese Area and Identification of New Docosahexaenoic Acid-Producing Species

**Mikihide Demura [1,\*], Seiji Noma [1,2] and Nobuyuki Hayashi [1,2]**

[1] Faculty of Agriculture, College of Natural Sciences, Institute of Education and Research, Saga University, Saga 840-8502, Japan; nomas@cc.saga-u.ac.jp (S.N.); yurika@cc.saga-u.ac.jp (N.H.)

[2] The United Graduate School, Agricultural Sciences, Kagoshima University, Kagoshima 890-0065, Japan

\* Correspondence: st8148@cc.saga-u.ac.jp

**Abstract:** *Desmodesmus* is a green microalgal genus that is frequently found in aquatic environments. Its high biomass productivity and potential as a source of lipids have attracted considerable attention. Although *Desmodesmus* is ubiquitous, it is difficult to identify; even within a small area, the diversity of the species and the fatty acids they produce are unknown. In this study, we performed scanning electron microscopy (SEM) and analyzed the genetic diversity of the internal transcribed spacer (ITS) region to accurately identify *Desmodesmus* in a local area of Japan (Saga City, Saga Pref.) and to assess its existence as a biological resource. In addition, we analyzed the fatty acid composition and content of the newly established strains. In total, 10 new strains were established, and 9 previously described species were identified. The presence of a cosmopolitan species indicated the global distribution of *Desmodesmus*. However, only regional species were found. One strain, dSgDes-b, did not form a clear clade with any described species in the phylogenetic analysis and had a characteristic ITS2 secondary structure. The cell wall of this strain exhibited a distinctive microstructure, and it produced docosahexaenoic acid (DHA); hence, the strain was described as a new species, *Desmodesmus dohacommunis* Demura sp. nov. Thus, useful information regarding the use of *Desmodesmus* as a bioresource was provided.

**Keywords:** *Desmodesmus*; diversity; docosahexaenoic acid (DHA); fatty acid; lipid; new species

## 1. Introduction

*Desmodesmus* (Chodat) An, Friedl, and Hegewald is a green algal genus that is frequently found in aquatic environments. Because of the ease of culturing and high biomass productivity, it has attracted attention as a material for biofuels [1,2]. In addition, the high lipid content indicates its potential as a source of high value-added fatty acids, such as ω-3 fatty acids [3]. Sijil et al. [3] reported that high α-linolenic acid production in a strain of *Desmodesmus* is enhanced by culturing under mixotrophic and low-temperature conditions. Furthermore, owing to its ability to remove nutrients, such as nitrogen and phosphorus, from wastewater while producing biomass, it is important from the perspective of resource recycling [4,5].

An et al. [6] established the genus *Desmodesmus* to include some members of the genus *Scenedesmus* that possess a unique cell surface microstructure and form a unique lineage in the family Scenedesmaceae. Subsequently, there have been numerous papers describing the recombination of species and establishment of new species [7–9]. However, identification using optical microscopy is difficult, and detailed studies using scanning electron microscopy (SEM) and DNA sequencing are necessary to accurately identify and determine the actual existence of any species of *Desmodesmus*. Currently, we have no knowledge regarding the community structure of the *Desmodesmus* species, in general, and the distribution or validity of certain species in the genus.

The aim of the present study was to determine the fatty acid composition of *Desmodesmus* strains and thereby find local strains that have potential for commercial applications. We identified newly established *Desmodesmus* strains based on detailed morphological observations and phylogenetic analysis and revealed the FAME(fatty acid methyl esters) profile of these strains.

## 2. Materials and Methods

### 2.1. Strain Establishment

Surface water samples of ponds and creeks were collected in plastic bottles between September 2018 and January 2019 in Saga City, Saga Prefecture, Japan (Figure 1). The coenobia of *Desmodesmus* were isolated from the water sample using a micropipette [10] under an inverted light microscope (CKX53, Olympus, Tokyo, Japan). Each coenobium was incubated in 24-well plastic plates containing AF6 medium [11] at 25 °C under a 12 h light/12 h dark cycle with white fluorescent illumination (approximately 100 μmol photons m$^{-2}$s$^{-1}$). After two weeks of incubation, only those isolates that exhibited noticeable growth were transferred to test tubes (15 mL of AF6 medium). Ten isolates were established as unialgal cultures. Eight isolates, with the exception of dSgDes-0 and dSgDes-eco6, were axenic, and a B-V medium was used to determine if cultures were axenic, using the procedure described by Ichimura and Watanabe [12]. All established isolates were maintained under the same conditions as described above.

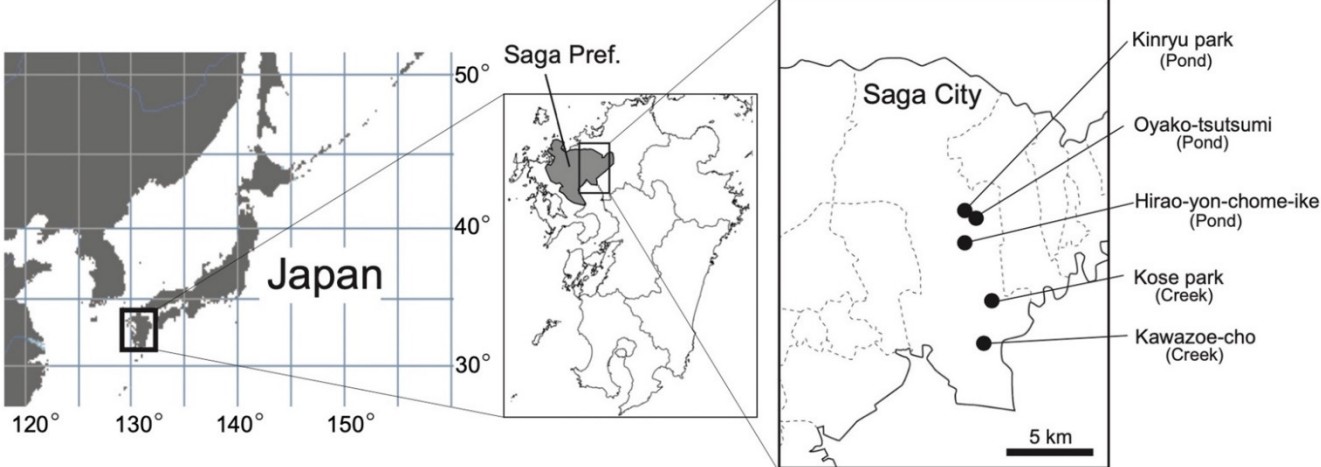

**Figure 1.** Sampling locations in Saga City, Saga Prefecture, Japan.

Images of the established isolates were acquired using a light microscope with a camera system (BX53 with DP74, Olympus, Tokyo, Japan). For cell size measurement of the isolate dSgDes-b, over 20 coenobia (4 cells per coenobium) were selected randomly, and cell size (length and width) was measured manually (Figure 2).

### 2.2. Scanning Electron Microscopy (SEM)

For SEM observation, sample preparation was based on the water freeze-drying method as follows [13,14]: The culture was fixed with 5% glutaraldehyde and centrifuged at 2000× *g* for 5 min. After removing the supernatant, deionized water was added, and the mixture was resuspended and recentrifuged. These operations were performed twice to remove glutaraldehyde. Finally, a drop of the pellet (ca. 10 μL) was placed on the sample table (diameter, 1 cm) of the SEM and frozen at −80 °C. After 24 h, the sample was dried in a vacuum dryer (EYELA FDU-2110, Tokyo RIKAKIKAI, Tokyo, Japan) and sputtered with gold (JEOL JFC-1600, Tokyo, Japan). SEM was performed using an SU-1500 (Hitachi, Tokyo, Japan).

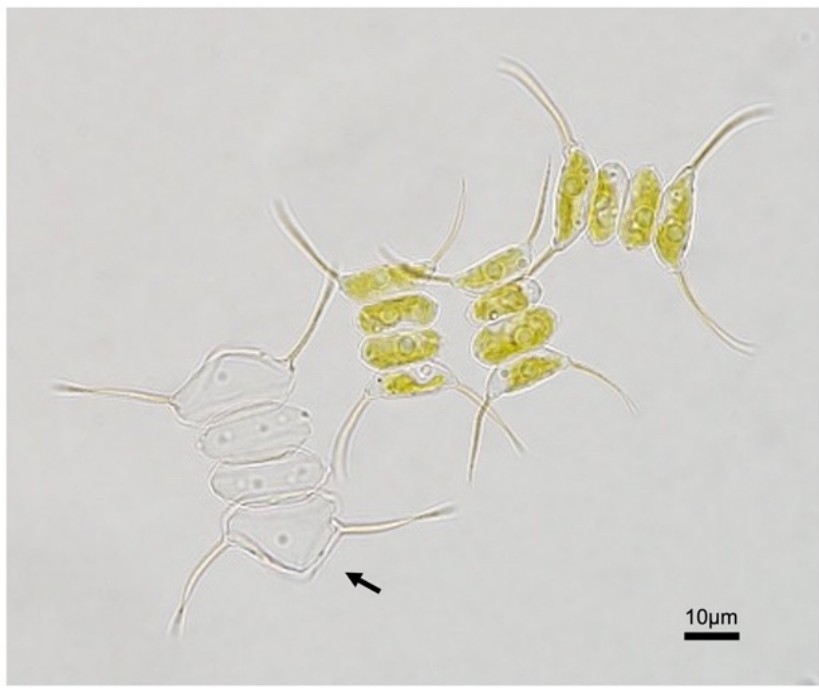

**Figure 2.** Light microscope photograph of *Desmodesmus dohacommunis* strain dSgDes-b. Arrow indicates an empty cell.

### 2.3. Phylogenetic Analysis

For extracting DNA from the 10 newly established isolates. The pellet obtained by centrifugation of the culture was used for DNA extraction. Total DNA was extracted using the DNeasy plant mini kit (Qiagen, Hilden, Germany) according to the manufacturer's protocol. Polymerase chain reaction (PCR) was performed in a TaKaRa thermal cycler (TaKaRa, Tokyo, Japan) in a 25 μL reaction mixture containing 0.2 mM of each dNTP, 10× PCR buffer, 0.25 U Ex Taq DNA polymerase (TaKaRa), 0.5 μM of each primer pair, and 0.5–5 ng DNA. A primer set, NS7m (5′-GGCAATAACAGGTCTGT-3′) and LR1850 (5′-CCTCACGGTACTTGTT-3′) [6], which was designed for the "ITS region", was used. The ITS region contains a part of the 18S ribosomal RNA gene, internal transcribed spacer 1(ITS1), 5.8S rRNA gene, ITS2, and a part of the 28S rRNA gene. The PCR conditions were as follows: 94 °C for 10 min; 30 cycles of 94 °C for 1 min, 50 °C for 45 s, and 72 °C for 1 min, and 72 °C for 5 min as final extension. The PCR product was purified using the QIAquick PCR purification kit (Qiagen, Hilden, Germany) according to the manufacturer's protocol. The sequence of the PCR product was determined by the Fasmac sequencing service (FASMAC, Atsugi, Japan).

Phylogenetic analyses were performed using MEGA version X [15]. For the analysis of only the ITS2 region (236 bp), 35 sequences, including 10 original sequences and 25 published sequences [8,9], were used. The sequences KU359289 (strain Hegewald 1977-141, *D. rectangularis*), KU359294 (strain Hegewald 1976-43, *D. pseudocommunis*), KU359279 (strain Hegewald 1974-35, *D. communis*), KU359302 (strain Hegewald 1977-2, *D. protuberans*), and KU359303 (strain Hegewald 1981-51, *D. pseudoprotuberans*) were those of the type strains identified by Hegewald and Braband [9].

The sequences were aligned using the "ClustalW" mode in MEGA X software with default settings. The "Models" mode was used to find the best substitution model. K2 + G was the best model for the ITS2 region (236 bp). "Construct/Neighbor-Joining (NJ) Tree" and "Construct/Maximum Likelihood (ML) tree" models were used to construct the phylogenetic tree with bootstrap number 100.

## 2.4. Comparison of RNA Secondary Structure

To construct the secondary structure of the ITS2 RNA, which is important for species identification of *Desmodesmus* [6,9], the UNAFolding web server [16] was used with default settings. The sequences of the type strain, i.e., KU359289 (*D. rectangularis*), KU359294 (*D. pseudocommunis*), KU359279 (*D. communis*), KU359302 (*D. protuberans*), and KU359303 (*D. pseudoprotuberans*), and the sequence of the isolate dSgDes-b were analyzed. The structures with the lowest ΔG (Gibbs free energy change)value were selected.

## 2.5. Fatty Acid Analysis

The 10 established isolates were incubated in a 250 mL flask containing 200 mL culture (185 mL of new AF6 medium and 15 mL culture) at 25 °C under a 12 h light/12 h dark cycle with white fluorescent illumination (approximately 100 μmol photons m$^{-2}$s$^{-1}$) and air bubbling at 150 mL/min for 15 days (three flasks per strain). After incubation, all culture samples (200 mL) were filtered onto a 1.0 μm pore glass filter (GF/C, ADVANTEC, Tokyo, Japan), and the filter was immediately dried by heating at 80 °C for 2 h. The dried filter was stored at −30 °C until the fatty acid methyl esters (FAMEs) were prepared and extracted.

FAMEs were prepared and extracted following a previous procedure [17]. The filter sample was transferred to a test tube (12 mL), and 5 mL of 3.6% (*w/w*) HCl methanol was added to 50 μL of C26:0 (1 mg/mL, methyl hexacosanoate, Sigma-Aldrich, Bellefonte, PA, USA) as an internal standard. The sample was then incubated in a heat block at 80 °C for 3 h. The test tube was gently stirred every hour. After cooling to room temperature, hexane (5 mL) was added, and the mixture was vortexed. After 15 min, the upper layer was transferred to a new test tube and evaporated with nitrogen gas. Finally, the extract was suspended in 100 μL hexane.

The FAMEs were analyzed using gas chromatography–mass spectrometry (GC-MS) using a Shimazdzu GC-MS 2010 (Shimadzu Co., Kyoto, Japan) equipped with a capillary column (DB-WAX, 0.25 mm i.d. × 30 m, film thickness 0.25 μm: Agilent, Santa Clara, CA, USA). The column temperature was programmed to increase from 160 °C to 260 °C at 4 °C/min. The temperature of the injection port was 250 °C. The FAMEs were analyzed and quantified based on their peak areas in the chromatogram relative to the peak area of the internal standard. The FAMEs were identified by comparing the retention times and fragment peak patterns with those of the Supelco 37 Component FAME mix (Supelco, ASIGMA-ALDRICH, St. Louis, MO, USA).

The content (%) of each identified FAME per dry weight of microalgae was calculated from the weight of the microalgae trapped on the filter and the amount of FAME present. The total lipid content (%) per dry weight was calculated from the sum of the amount of identifiable FAMEs and that of other lipids (unidentifiable FAMEs and hydrocarbon). The amount of fatty acids identified was set at 100%, and the percentages of saturated fatty acid (SFA), monounsaturated fatty acid (MUFA), and polyunsaturated fatty acid (PUFA) were calculated.

To estimate the similarities in FAME composition and content among the strains, principal component analysis (PCA) was performed using the default settings of SPSS software, version 26 (IBM, Armonk, WA, USA).

## 3. Results

### 3.1. Identification of Local Desmodesmus

In total, 10 isolates were established from Japanese local areas in Saga City, Saga Prefecture. With the exception of isolate dSgDes-b, these isolates could be identified as previously recorded species, namely, *D. armatus*, *D. communis*, *D. pirkollei*, *D. protuberans*, and *D. tropicus*, on the basis of morphological observations and molecular phylogenetic analysis (Table 1 and Figure 3).

**Table 1.** List of strains analyzed.

| Strain Name | Species of *Desmodesmus* | Locality in Saga City | Sampling Day | Accession No. |
| --- | --- | --- | --- | --- |
| dSgDes-i | *D. armatus* (Chodat) E. Hegewald | Kawazoe-cho (Creek) | 21 August 2018 | LC642118 |
| dSgDes-m | | Kinryu park (Pond) | 9 July 2019 | LC642119 |
| dSgDes-eco1 | *D. communis* (E. Hegewald) E. Hegewald | Hirao-yon-chome-ike (Pond) | 23 January 2020 | LC642120 |
| dSgDes-eco11 | | | 17 March 2020 | LC642121 |
| dSgDes-b | *D. dohacommunis* Demura sp. nov. | Oyako-tsutsumi (Pond) | 28 November 2018 | LC642122 |
| dSgDes-0 | *D. pirkollei* E. Hegewald | Hirao-yon-chome-ike (Pond) | 15 June 2017 | LC642123 |
| dSgDes-eco12 | | | 17 March 2020 | LC642124 |
| dSgBigDeka4 | *D. protuberans* (F.E.Fritsch & M.F.Rich) E.Hegewald | Kose park (Creek) | 21 April 2020 | LC642125 |
| dSgDes-eco6 | *D. tropicus* (W.B.Crow) E. Hegewald | Hirao-yon-chome-ike (Pond) | 13 February 2020 | LC642126 |
| dSgBigDeka1 | | Kose park (Creek) | 21 April 2020 | LC642127 |

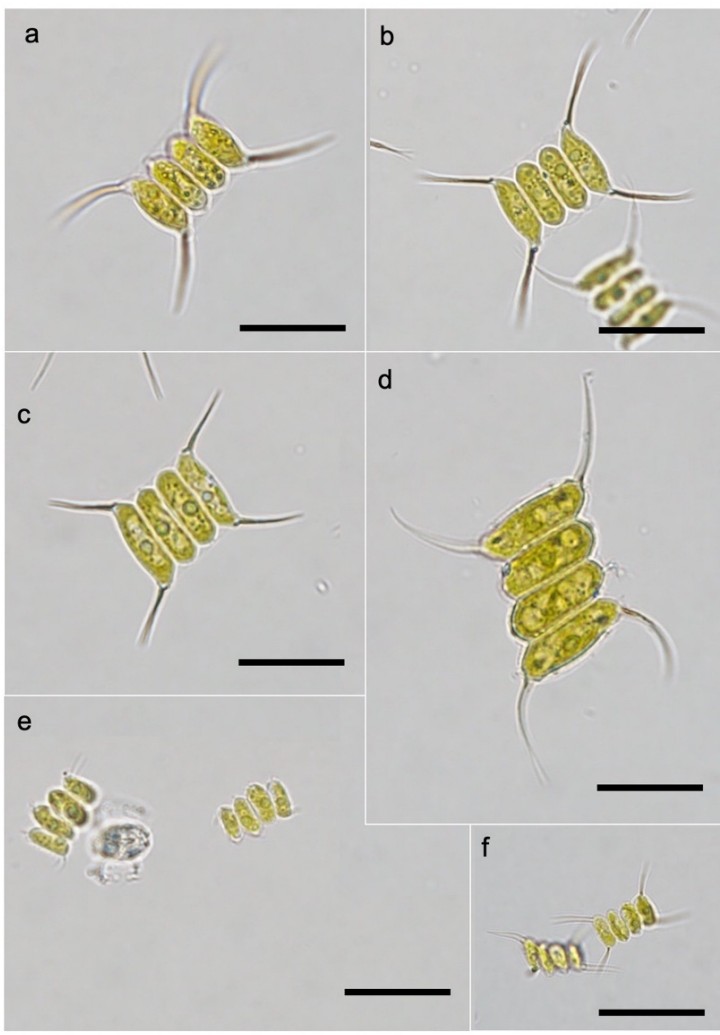

**Figure 3.** Light microscope photographs of *Desmodesmus* species established in this study: *Desmodesmus communis* strains dSgDes-eco1 (**a**), dSgDes-eco11 (**b**), *D. protuberans* strain dSgBigDeka4 (**c**), *D. tropicus* strain dSgDes-eco6 (**d**), *D. pirkollei* strain dSgDes-eco12 (**e**), and *D. armatus* strain dSgDes-m (**f**). Scale bars = 20 μm.

All strains had spines, which is characteristic of *Desmodesmus*. Four strains, dSgDes-eco1, dSgDes-eco11, dSgDes-b, and dSgBigDeka4, were indistinguishable under the optical microscope (Figures 2 and 3). However, an observation of the cell surface microstructure using SEM showed that the two groups could be distinguished. The rosettes of three strains, dSgDes-eco1, dSgDes-eco11, and dSgBigDeka4, were usually composed of approximately four–five tubes touching at the center or contained an additional central tube; the net-like cell wall had irregular meshes and a large mesh size, which is characteristic of *D. communis* and *D. protuberans* [9] (Figure 4a,b). In contrast, the rosettes of the strain dSgDes-b were complex and had a large number of outer tubes (eight–nine); although rarely, rosettes with four–five tubes were observed. The net-like structure of the outer cell wall layer had regular and smaller meshes than the three strains mentioned above (Figure 4c). Two strains, dSgDes-eco1 and dSgDes-eco11, and the strain dSgBigDeka4 were distinguished by the differences in the ITS2 sequence. dSgDes-eco1 and dSgDes-eco11 were identified as *D. communis* and dSgBigDeka4 as *D. protuberans*, according to Hegewald and Braband [9].

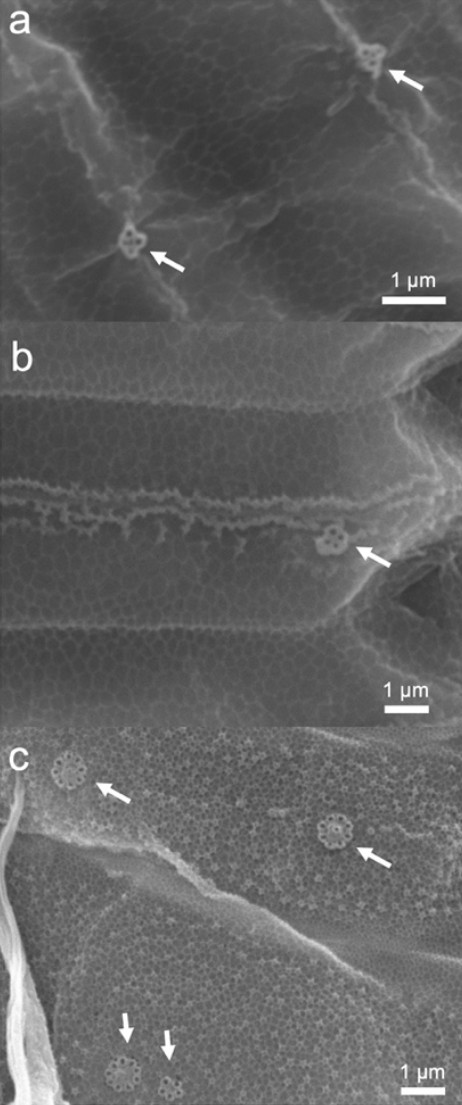

**Figure 4.** Scanning electron micrographs of *D. communis* strain dSgDes-eco1 (**a**), *D. protuberans* strain dSgBigDeka4 (**b**), and *D. dohacommunis* strain dSgDes-b (**c**). Arrow indicates the rosette on the cell wall.

The strain dSgDes-eco6 was clearly larger than the other strains (Figure 3d), and in some cases, the coenobium with holes (small space) between cells showed characteristics of

*D. tropicus* [7]. The strains dSgDes-0 and dSgDes-eco12 showed a large number of shorter spines not only on the outer cells but also on the inner cells, which is a characteristic of *D. pirkollei* [18] (Figure 3e). The strains dSgDes-i and dSgDes-m had smaller coenobium with outer cell spines (Figure 3f). When observed using SEM, obvious ridge decoration of the cell wall (data not shown), which is a characteristic of *D. armatus*, was recognized [19].

Some species coexisted in the same pond on the same day, and some species were distributed across multiple ponds (Table 1).

### 3.2. Genetic Diversity of Local Desmodesmus

The two molecular phylogenic trees (NJ and ML) showed the same pattern. A phylogenetic tree of the ITS2 region (236 bp) for 35 sequences, including 10 original sequences, is shown in Figure 5. Clades B–F were supported by high bootstrap values, each corresponding to one species (clade B, *D. pseudocommunis*; clade C, *D. maximus*; clade D, *D. tropicus*; clade E, *D. pirkollei*; clade F, *D. armatus*). Clade A was a mixture of several *D. communis*-related species. The sequences of strains dSgDes-eco1 and dSgDes-eco11 were consistent with those of the type strain of *D. communis*. The sequences of strain dSgBigDeka4 and *D. protuberans* formed a clade that supported relatively high bootstrap values. The sequences of strain dSgDes-b did not show a strong genetic relationship with any of the sequences in clade A.

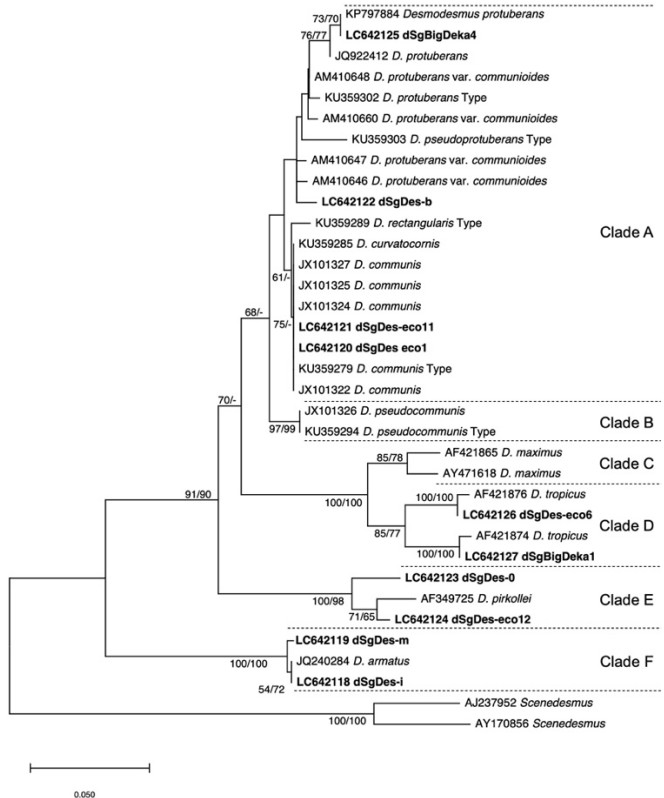

**Figure 5.** Neighbor-joining tree based on ITS2 region sequences. Bootstrap values > 50 are shown near branches (NJ/ML).

### 3.3. Comparison of RNA Structure

RNA structure and compensatory base changes (CBCs) are important for the identification of *Desmodesmus* [6,9]. CBC is a base mutation that occurs in both nucleotides of a paired structural position [20]. Hemi-CBC is a mutation of a single nucleotide in a paired structural position while the nucleotide bond is retained. A comparison of the RNA secondary structure of the five type strains of clade A and B (Figure 5) revealed multiple hemi-CBCs and differences in loop structure in helices I, II, and IV (Figure 6). The "G-U" at

the bottom of loop no. 15–19 of helix I and the loop structure of helix IV were detected only in strain dSgDes-b.

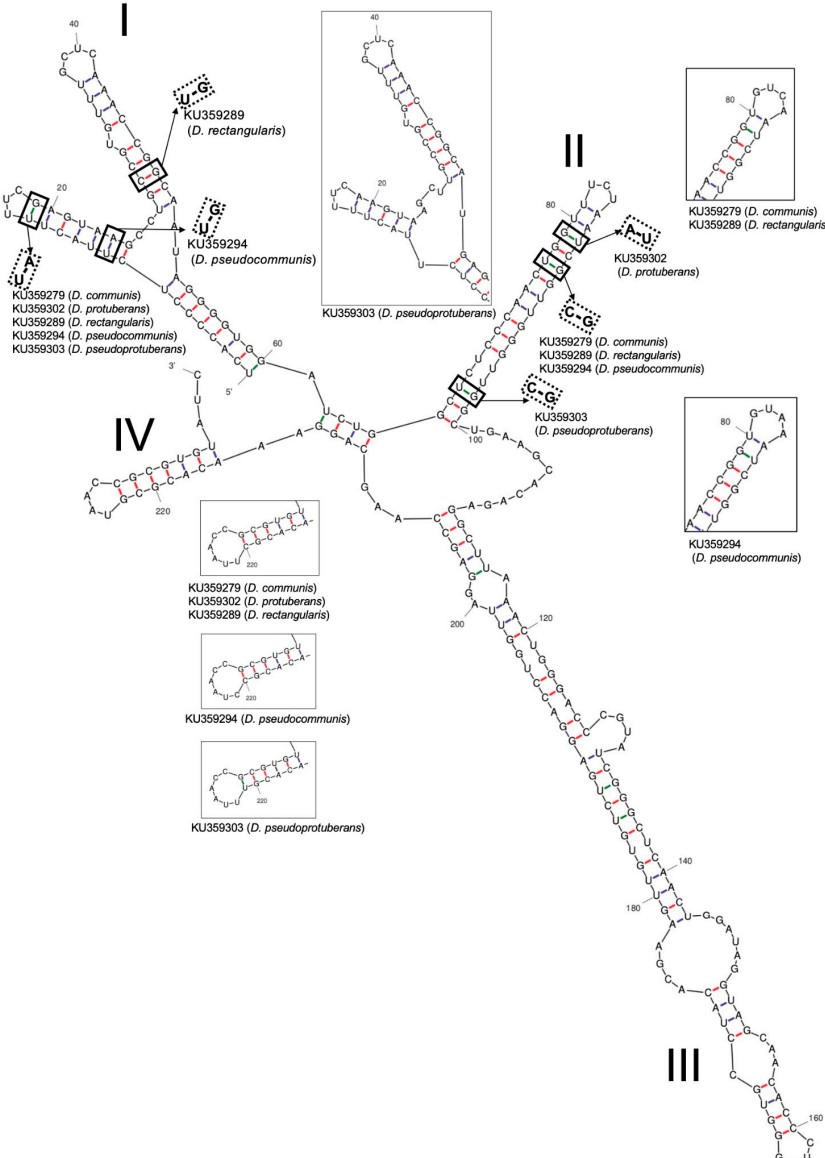

**Figure 6.** Differences in ITS2 base change and RNA secondary structure between *D. dohacommunis* strain dSgDes-b and five species. The sequences of the five species were determined from type strains in Hegewald and Braband [9].

### 3.4. Diversity of FAMEs in Local Desmodesmus

The composition and content of fatty acids in the 10 strains are listed in Table 2. α-Linolenic acid (18:3(n-3)) was abundant in all strains, followed by palmitic acid (16:0). There was a difference in the third most abundant fatty acid; dSgDes-m, dSgDes-i, dSgDes-eco1, dSgDes-eco11, dSgDes-0, and dSgBigDeka1 had linoleic acid (18:2(n-6)), whereas dSgDes-b, dSgDes-eco12, dSgBigDeka4, and dSgDes-eco6 had oleic acid (18:1(n-9)). DHA was detected only in the dSgDes-b strain (0.014 ± 0.003%). Among the fatty acids that were identified, all strains had a high PUFA content (Figure 7). The SFA content of dSgBigDeka1 was approximately 1.5 times higher than that of dSgDes-i. The strains dSgDes-b and dSgDes-eo6 contained more MUFA than the other strains. The maximum total lipid content was 12.5% for dSgDes-m and the minimum was 6.7% for dSgDes-b (Table 2).

**Table 2.** Fatty acid composition of 10 strains (%). n.d: not detected.

| Species | *D. armatus* | | *D. communis* | | *D. dohacommunis* | *D. pirkollei* | | *D. protuberans* | *D. tropicus* | |
| --- | --- | --- | --- | --- | --- | --- | --- | --- | --- | --- |
| Strains | dSgDes-i | dSgDes-m | dSgDes-eco1 | dSgDes-eco11 | dSgDes-b | dSgDes-0 | dSgDes-eco12 | dSgBigDeka4 | dSgDes-eco6 | dSgBigDeka1 |
| C12 | n.d | n.d | 0.00 ± 0.00 | n.d | n.d | 0.01 ± 0.00 | 0.00 ± 0.00 | 0.00 ± 0.00 | 0.00 ± 0.00 | 0.00 ± 0.00 |
| C15 | n.d | 0.00 ± 0.00 | 0.00 ± 0.00 | 0.00 ± 0.00 | 0.00 ± 0.00 | n.d | 0.02 ± 0.02 | 0.01 ± 0.00 | 0.00 ± 0.00 | 0.00 ± 0.00 |
| C16 | 1.00 ± 0.02 | 1.49 ± 0.09 | 1.29 ± 0.06 | 1.37 ± 0.08 | 0.83 ± 0.14 | 0.91 ± 0.07 | 1.27 ± 0.15 | 1.22 ± 0.02 | 1.37 ± 0.21 | 1.36 ± 0.03 |
| 16:2(n-4) | 0.14 ± 0.00 | 0.25 ± 0.02 | 0.17 ± 0.03 | 0.22 ± 0.01 | 0.11 ± 0.02 | 0.03 ± 0.00 | 0.05 ± 0.00 | 0.14 ± 0.00 | 0.06 ± 0.02 | 0.05 ± 0.00 |
| C17 | 0.00 ± 0.00 | n.d | 0.00 ± 0.00 | n.d | 0.00 ± 0.00 | n.d | 0.00 ± 0.00 | 0.00 ± 0.00 | 0.00 ± 0.00 | n.d |
| 16:3(n-4) | 0.21 ± 0.01 | 0.32 ± 0.02 | 0.12 ± 0.02 | 0.10 ± 0.00 | 0.05 ± 0.01 | 0.11 ± 0.01 | 0.14 ± 0.01 | 0.13 ± 0.00 | 0.21 ± 0.03 | 0.07 ± 0.00 |
| C18 | 0.02 ± 0.00 | 0.03 ± 0.00 | 0.02 ± 0.00 | 0.02 ± 0.00 | 0.02 ± 0.00 | 0.03 ± 0.00 | 0.04 ± 0.00 | 0.04 ± 0.00 | 0.05 ± 0.00 | 0.05 ± 0.00 |
| 18:1(n-9) | 0.53 ± 0.06 | 0.90 ± 0.03 | 0.66 ± 0.03 | 0.65 ± 0.04 | 0.80 ± 0.04 | 0.32 ± 0.03 | 0.71 ± 0.07 | 0.64 ± 0.00 | 1.08 ± 0.14 | 0.59 ± 0.03 |
| 18:1(n-7) | 0.03 ± 0.01 | 0.07 ± 0.00 | 0.10 ± 0.02 | 0.07 ± 0.00 | 0.07 ± 0.00 | 0.31 ± 0.04 | 0.05 ± 0.00 | 0.06 ± 0.00 | 0.12 ± 0.06 | 0.04 ± 0.00 |
| 18:2(n-6) | 0.66 ± 0.01 | 1.00 ± 0.03 | 0.70 ± 0.04 | 0.75 ± 0.04 | 0.51 ± 0.06 | 0.33 ± 0.02 | 0.43 ± 0.04 | 0.61 ± 0.01 | 0.51 ± 0.04 | 0.59 ± 0.02 |
| 18:3(n-6) | 0.05 ± 0.00 | 0.07 ± 0.00 | 0.15 ± 0.00 | 0.16 ± 0.01 | 0.06 ± 0.01 | 0.12 ± 0.14 | 0.04 ± 0.00 | 0.05 ± 0.00 | 0.08 ± 0.01 | 0.05 ± 0.00 |
| 18:3(n-3) | 2.06 ± 0.10 | 2.75 ± 0.09 | 2.36 ± 0.14 | 2.52 ± 0.15 | 1.26 ± 0.25 | 1.91 ± 0.15 | 2.24 ± 0.25 | 2.22 ± 0.02 | 1.76 ± 0.22 | 1.83 ± 0.08 |
| 18:4(n-3) | 0.11 ± 0.07 | 0.21 ± 0.01 | 0.15 ± 0.00 | 0.16 ± 0.01 | 0.19 ± 0.03 | 0.11 ± 0.02 | 0.21 ± 0.02 | 0.18 ± 0.00 | 0.19 ± 0.04 | 0.13 ± 0.00 |
| C20 | 0.02 ± 0.00 | n.d | n.d | n.d | n.d | n.d | 0.00 ± 0.00 | n.d | n.d | n.d |
| 20:1(n-9) or (n-11) | 0.04 ± 0.00 | n.d | 0.00 ± 0.00 | 0.00 ± 0.00 | 0.02 ± 0.00 | 0.01 ± 0.00 | 0.01 ± 0.00 | 0.02 ± 0.00 | 0.08 ± 0.07 | 0.01 ± 0.01 |
| C22 | 0.03 ± 0.00 | 0.04 ± 0.00 | 0.03 ± 0.01 | 0.03 ± 0.00 | 0.02 ± 0.00 | 0.03 ± 0.00 | 0.03 ± 0.00 | 0.04 ± 0.00 | 0.04 ± 0.01 | 0.03 ± 0.01 |
| C24 | 0.03 ± 0.00 | n.d | 0.02 ± 0.00 | 0.02 ± 0.00 | 0.02 ± 0.00 | 0.02 ± 0.00 | 0.02 ± 0.00 | 0.02 ± 0.00 | 0.01 ± 0.01 | n.d |
| 22:6(n-3) | n.d | n.d | n.d | n.d | 0.01 ± 0.00 | n.d | n.d | n.d | n.d | n.d |
| Total Lipids | 8.25 ± 0.34 | 12.5 ± 1.29 | 9.76 ± 0.37 | 10.4 ± 0.42 | 6.71 ± 1.07 | 7.00 ± 0.48 | 8.60 ± 0.74 | 8.68 ± 0.14 | 9.26 ± 0.34 | 8.17 ± 0.15 |

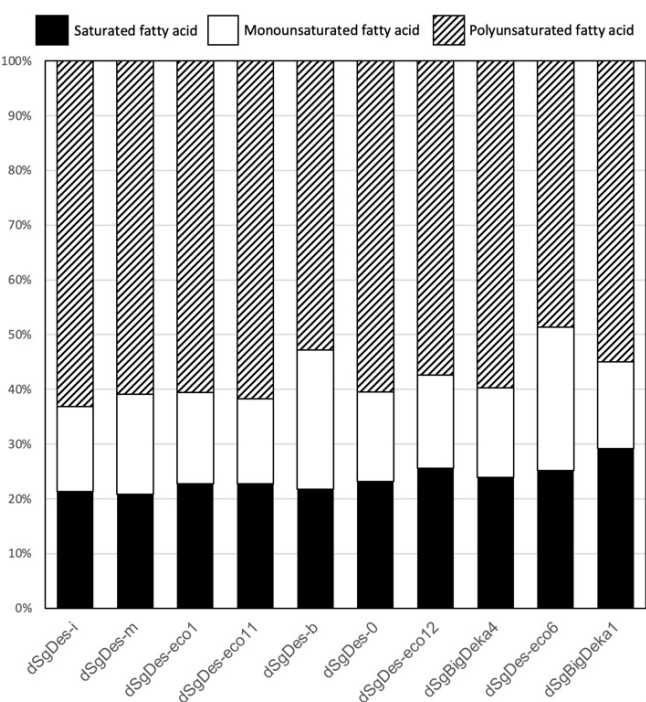

**Figure 7.** Percentage of saturated fatty acids, monounsaturated fatty acids, and polyunsaturated fatty acids among the identified fatty acids.

The PCA results showed that the first principal component was 97.6% and the second principal component was 1.5% (Figure 8). The species tended to aggregate, such as strains dSgDes-eco1 and dSgDes-eco11 (*D. communis*) and strains dSgDes-m and dSgDes-i (*D. armatus*). In contrast, the strains differed considerably in terms of fatty acid composition and content, even for strains belonging to the same species, such as dSgDes-eco6 and dSgBigDeka1 (*D. tropicus*) and strains dSgDes-eco12 and dSgDes-0 (*D. pirkollei*). Strain dSgDes-b clustered near the strain dSgDes-eco6.

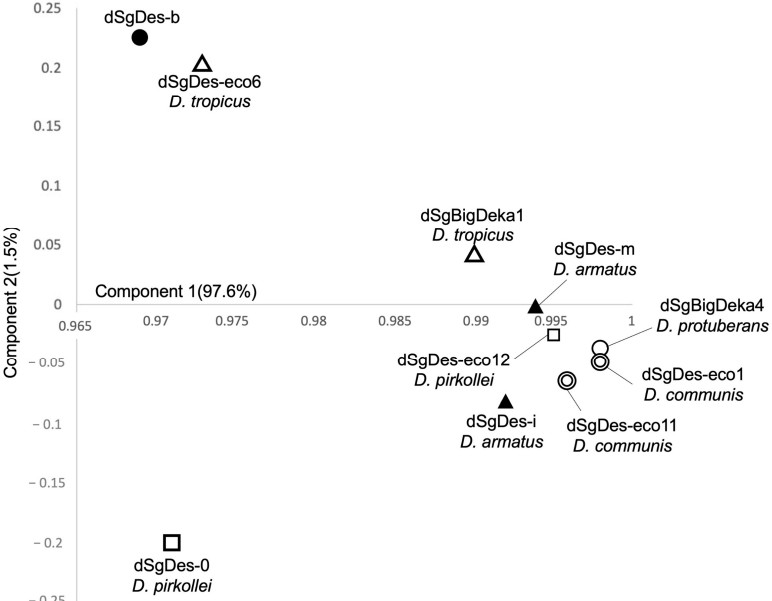

**Figure 8.** Principal component analysis (PCA) based on fatty acid content and composition data of the established strains in this study. Black circle: strain "dSgDes-b", white crircle: strain of *D. protuberans*, double circle: strains of *D. communis*, black triangle: strains of *D. armatus*, white triangles: strains of *D. tropicus*, square: strains of *D. pirkollei*.

## 4. Discussion

In this study, we were able to establish 10 new culture strains, of which 9 were identified to be 6 known species, and to reveal their fatty acid profiles. Strains dSgDes-i and dSgDes-m, dSgDes-0 and dSgDes-eco12, and dSgDes-eco6 and dSgBigDeka1 could be clearly identified as *D. armatus*, *D. pirkollei*, and *D. tropicus*, respectively, on the basis of morphological characteristics and phylogenetic analysis. Strains dSgDes-eco1 and dSgDes-11 and the strain dSgBigDeka4 could be identified as *D. communis* and *D. protuberans*, respectively, on the basis of phylogenetic analysis. For these two species, Hegewald and Braband [9] pointed out that they cannot be distinguished on the basis of morphology and that only the ITS2 sequence is the key to identification. In addition, we were able to establish a new culture strain, dSgDes-b, which did not belong to any known species. The cell surface microstructure of strain dSgDes-b was similar to that of *D. pseudocommunis* [9], but it was distinguished by the presence of rare four–five tube structures in the rosette (Figure 4c). The ITS2 sequence of the strain dSgDes-b was also similar to that of *D. communis* and five other species (clades A and B in Figure 5). Compared to those of other clade A and B strains, this strain had a unique secondary RNA structure (Figure 6). Studies have shown that the secondary structure of ITS2 and base substitutions, known as CBC (compensatory base changes) and hemi-CBC, reflect differences among biological species of microalgae (e.g., [20–23]). It is also one of the key characteristics of species recognition in *Desmodesmus* [6,9]. CBC was not detected between *D. communis*, *D. rectangularis*, *D. protuberans*, *D. pseudocommunis,* and *D. pseudoprotuberans*; however, hemi-CBC was detected between the species [9]. The difference in secondary structure and the presence of multiple hemi-CBCs also support the idea that strain dSgDes-b is an independent species.

Furthermore, fatty acid analysis of the strain dSgDes-b revealed the synthesis of DHA, although few studies have reported DHA production in *Desmodesmus* and the closely related *Scenedesmus* [24,25]. Lang et al. [24] reported the fatty acid profiles of 21 *Desmodesmus* strains and 29 *Scenedesmus* strains in the algae culture collection of Gottingen University (SAG); no strains were found to contain DHA. Soares et al. [25] indicated that *D. denticulatus* produced 0.1% DHA (content of dry weight). Although the content of strain dSgDes-b was $0.014 \pm 0.003$%, which is lower than that reported for *D. denticulatus*, it may be a valuable *Desmodesmus* strain that may act as a biological resource. The percentage of unsaturated fatty acids increases when microalgae are cultured at low temperatures [26,27]. In future, the culture conditions that can increase DHA content in dSgDes-b have to be investigated.

Although dSgDes-b is unique in producing DHA, the composition and content of other fatty acids were similar to those of dSgDes-eco6 (Figure 8). The correlation between fatty acid composition and content (Figure 8) did not clearly reflect the relationship observed in the phylogenetic analysis (Figure 5). Lang et al. [24] determined the fatty acid profiles of more than 2000 cultured strains in the SAG collection, which included a large number of taxa, and pointed out that the similarity in the fatty acid profiles is evident at the phylum and class levels; hence, we believe that identification of species clustering at the genus level was difficult in this study. The common feature of high oleic and palmitic acid content was also consistent with the previously reported characteristics of *Desmodesmus* [1–3]. Because all strains were high in PUFAs, they would be more useful as ω-3 fatty acids for dietary supplement applications rather than for fuel applications, as pointed out by Sijil et al. [3].

There were differences in fatty acid composition between the strains. The strain dSgBigDeka1 is relatively high in SFAs, whereas dSgDes-b and dSgDes-eco6 are relatively high in MUFAs. By selecting the appropriate culture strain, it is possible to produce fatty acids as required. In particular, dSgDes-b may be of use for DHA production.

Diversity in total lipid production was also observed between the strains. Reportedly, the total lipid content of *Desmodesmus* exceeds 30% in some cases, but all results were obtained under stress conditions, such as wastewater and nitrogen deficiency [2,28]. Hence, it is necessary to examine the total lipid production of the strains established in this study under stress conditions. In this case, dSgDes-m, which showed the maximum value, would be a promising strain for lipid production.

Six species of *Desmodesmus* were recognized from only five water bodies in a very small area of Japan. The coexistence of multiple species in the same pond was also detected; e.g., *D. communis*, *D. pirkollei*, and *D. tropicus* were identified in a single pond (Hirao-yon-chome-ike). In a study of *Desmodesmus* diversity in lakes in Minnesota, USA, a large number of species from a small area were sympatric [29]. Owing to the high diversity of *Desmodesmus,* rich biological resources may be available even in a limited area.

The fact that the same species was observed in different ponds may indicate that the dispersal of *Desmodesmus* occurs frequently. *D. communis* has been identified in Europe, Russia, Asia, South America, and North America [9]; in addition, this species was also found to be distributed in Japan in this study. As dSgDes-eco1 and dSgDes-eco11 in this study showed the same ITS2 sequence as the previously reported type strain (Hegewald 1974-35, Finland), it is possible that global distribution diffusion and genetic homogenization are occurring constantly. Finlay [30] suggested that most microbial species show a global distribution, and even those that are considered endemic to a region may find new habitats via extensive sampling. *D. pirkollei* was identified as a species based on a strain isolated from a lake in Bali, which is described as a "very rare species" by Hegewald et al. [18]; however, in this study, it was found outside Bali, which supports the observations of Finlay [30]. Two possible reasons may explain why some species, such as *D. communis*, are identified globally, whereas others, such as *D. dohacommunis* and *D. pirkollei*, are rarely identified. The first possibility is that *D. dohacommunis* and *D. pirkollei* are relatively recently differentiated species that have not spread for a long time; the second possibility is that *D. dohacommunis* and *D. pirkollei* are old differentiated species that were once widely distributed but are now extinct in many places and survive only in limited environments. As *Desmodesmus* is known to be plastic in nature [31–34], it possibly has the ability to adapt and evolve its morphology and physiological activities in various ways depending on the environmental conditions. Although the feasibility of the above possibilities has not yet been ascertained, results indicate that the distribution and diffusion abilities of *Desmodesmus* are high. It is possible that the *D. dohacommunis* described here will also be found in other parts of the world in the future.

This study revealed genetic and fatty acid diversities in *Desmodesmus*, thus suggesting that commercially available *Desmodesmus* strains exist within small areas.

Description of new taxa: *Desmodesmus dohacommunis* Demura sp. nov.

Cell size was 9.4–15.8 × 3.2–4.7 μm. Coenobia of four cells were observed, outer cells of which had spines. This strain could not be distinguished from *D. communis* and its related species using optical microscopy. Rosettes of this strain were complex and had a large number of outer tubes (eight–nine), similar to those of *D. pseudocommunis*; however, rarely, rosettes of four–five tubes were observed. The net-like structure of the outer cell wall layer had regular and smaller meshes than those of *D. communis*. The sequences and the secondary structure of the ITS2 region distinguish the new species from *D. communis* and *D. pseudocommunis.* The details of this strain are mentioned below:

GenBank accession number: LC642122
Holotype: Shown in Figure 2
Type strain: dSgDes-b
Type locality: Oyako-tutumi, Saga City, Saga Prefecture, Japan (N33.326, E130.312)
Habitat: Pond
Etymology: The specific epithet "dohacommunis" is a combination of a contraction of the fatty acid <u>d</u>ocosa<u>h</u>exaenoic <u>a</u>cid ("doha"), which is present in the new species, and "communis", considering the similar morphological characteristics of the new species and the existing species *D. communis* and *D. pseudocommunis.*

## 5. Conclusions

In this study, we established 10 strains of 5 described species and a new species from a small Japanese area. It was found that multiple species of *Desmodesmus* coexisted even in

this small area. In addition, a cosmopolitan species reported in other parts of the world was found to be mixed with the species reported only in the limited area of this study. The 10 strains showed not only some similarities but also some diversities in fatty acid composition and content. A new species, *D. dohacommunis*, produced DHA, a highly valuable fatty acid. This species possessed original microstructures on the cell surface and a unique ITS sequence, and it could be distinguished from the existing species. We reveal that the diversity of *Desmodesmus* was very high even in a small area, indicating that it was a potential biological resource.

**Author Contributions:** Conceptualization, M.D.; methodology, M.D.; investigation, M.D.; resources, M.D.; writing—original draft preparation, M.D.; writing—review and editing, M.D., S.N. and N.H. All authors have read and agreed to the published version of the manuscript.

**Funding:** The authors are grateful to the Biomass Industry Promotion Division, Saga-City, for supporting this research. Part of the FAME analysis was conducted at Analytical Research Center for Experimental Sciences, Saga University, supported by "Project for Promoting Public Utilization of Advanced Research Infrastructure". The rest of the FAME analysis and the SEM observation were the result of using research equipment shared in MEXT project for promoting public utilization of advanced research infrastructure (program for supporting introduction of the new sharing system), Grant Number JPMXS0422400020 and JPMXS0422400021.

**Institutional Review Board Statement:** Not applicable.

**Informed Consent Statement:** Not applicable.

**Data Availability Statement:** The datasets generated and analyzed during the current study are available from the corresponding author on reasonable request.

**Conflicts of Interest:** The authors declare no conflict of interest.

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
