# Peer review of "Species and Fatty Acid Diversity of Desmodesmus (Chlorophyta) in a Local Japanese Area and Identification of New Docosahexaenoic Acid-Producing Species"

_2673-8783, doi:10.3390/biomass1020008_

Round 1

Reviewer 1 Report

My comments are attached in the file.

Author Response

Response to Reviewer’s Comments

We have seriously considered all the comments and carefully revised the manuscript. Our responses to the comments by the two reviewers are listed below item by item. We thank the two reviewers for their highly valuables comments and corrections, helping us to improve our manuscript.

Reviewer #1
An abstract needs to be changed and some sentences should be the passive voice. line 20, “we also found only regional species”. It can be changed into “only regional species were found” or something similar. 
Line 23, “we described it as new species” should be modified.
Line 24, “we were able to”, change this as well.
Response – Thank you for your comments. We modified our manuscript according to your advice in the revised abstract (L.20-25).

Keywords, the first word should be capital
Response – We revised our Keywords (L.26).

Line 31, “material for biofuels [e.g., 1,2], “e.g.,” is not necessary with these references. 
Response – We revised our manuscript (L.30).

Line 47 “the aim of the project” ?? is this a project or study? Change this line 
Line 48, the repetition sentence is used “in this study, the aim was to identify”. Change this sentence to “the research aims to identify” or similar wording. 
Line 50, “analysis and to reveal” remove the word “to” 
Response – Thank you for your valuable comments. We modified our manuscript according to your advice as follows (L.47-50).

The aim of the study was to determine the fatty acid composition of Desmodesmus strains and thereby find local strains that have potential for commercial applications. We identified the newly established Desmodesmus strains based on detailed morphological observations and phylogenetic analysis and revealed the FAME profile of these strains.

Line 74, (section 2.2) use Scanning electron microscopy (SEM) 
Response – We revised our manuscript (L.86).

Line 81, Change “The next day” to “after 24 hours” 
Response – We revised our manuscript (L.92-93).

Line 82, SEM was performed using Hitachi SU-1500? Means? 
Response – We revised our manuscript (L.95).

Section 2.3. is too long and complex to understand. Use concise sentences. 
Response – We followed your advice and made a few sentences more concise (L.98-139).

Line 133, FAMEs were prepared and extracted... change this line to “FAMEs were prepared and extracted following the previous literature [17]. 
Response – We revised our manuscript (L.161).

Line 162 and 163 are one heading or separate? 
Response –The section “Description of new taxa” was moved to the end and included in the discussion (L.503-520).

Line 164 in section 3.1, is not described well, the authors did not refer to the size with any figure here. It should be, “the size of 3.2-4.7 um was measured under SEM or OM etc” or similar. 
Response – Thank you for your comments. In the method section, there is a description as follows. For clarity, we have added a quote “Figure 2”(L.79-81).
For cell size measurement of the isolate dSgDes-b, over 20 coenobia (4 cells per coenobium) were selected randomly, and cell size (length and width) was measured manually (Figure 2).

Line 184, the caption of figure 2 is not well defined. How was it measured? 
Response – Thanks a lot for this indication. And sorry for this mistake. The legend in Figure 2 has been corrected (L.274-275). 

Line 231-234 should be written concisely; many words are repeated, that confuse the reader. 
Response – Thank you for your valuable comments. We have revised this paragraph concisely as follows (L.316-317).
Some species coexisted in the same pond on the same day, and some species were distributed across multiple ponds (Table 1).

In line 319 – end of the first sentence, a yellow highlight can be seen easily. 
Response – We revised our manuscript (L.414).

Link line 340 with 341.
Response – We revised our manuscript as follows (L.435-436).
Furthermore, fatty acid analysis of the strain dSgDes-b revealed synthesis of DHA, although few studies have reported DHA production in Desmodesmus and the closely related Scenedesmus [24,25].

Rephrase lines 402 – 404 into a single sentence.
Response – We revised our manuscript as follows (L.500-501).

This study revealed that genetic and fatty acid diversities in Desmodesmus, thus suggesting that commercially available Desmodesmus strains exist within such small areas.

Line 409, change the line “we found that” to “It was found or observed” 
Response – We revised our manuscript (L.526).

Line 415-416, “This study reveals that” should be changed as it repeats several times in conclusion and paper. 
Response – We revised our manuscript (L.532).

Reviewer 2 Report

  • Please revise the highlighted sentences and comments in the attached file.
  • The results section must be modified, taking into account the sequence of work and results: isolation, identification, and characterization.

Author Response

Response to Reviewer’s Comments

We have seriously considered all the comments and carefully revised the manuscript. Our responses to the comments by the two reviewers are listed below item by item. We thank the two reviewers for their highly valuables comments and corrections, helping us to improve our manuscript.

Reviewer #2
Line 11, microalgal 
Response – We revised our manuscript (L.11).

Line 65 ,isolates not strains ..after identification you can use the term strains
Response – We revised our manuscript.

Line 126, ?
Response – Thank you for this remark. We revised our manuscript (L.153-154).

Line 134, or: Anhydrous methanolic 5% (w/v) HCl. Please revise the reference No. 17
Response –Sorry we missed. We revised our manuscript (L.162).

Line 162, I think mentioning "new taxa" before comparing sequences is a prejudgment"
Response – Thank you for your valuable comments. The section “Description of new taxa” was moved to the end and included in the discussion (L.503-520).

Line 197, It is preferable to use a standard size for pictures, or to put the scale with each picture
Response –We pit the scale with each picture (Figure 3).

Line 258, ref.
Response –We added a reference (L. 352).

Line 296, try to combine the two parts in one table.
Response –We tried to combine Table 2, but the numbers were so small that we split it unavoidably.
